



# Description and evaluation of the new UM-UKCA (vn11.0) Double Extended Stratospheric-Tropospheric (DEST vn1.0) scheme for comprehensive modelling of halogen chemistry in the stratosphere.

Ewa M. Bednarz[1,a], Ryan Hossaini[1,2], Luke Abraham[3,4], and Martyn P. Chipperfield[5,6]

1. Lancaster Environment Centre, Lancaster University, Lancaster, UK
2. Centre of Excellence in Environmental Data Science, Lancaster University, Lancaster UK.
3. Department of Chemistry, University of Cambridge, Cambridge, UK
4. National Centre for Atmospheric Science (NCAS), UK
5. School of Earth and Environment, University of Leeds, Leeds, UK
6. National Centre for Earth Observation (NCEO), University of Leeds, Leeds, UK
 a. now at: CIRES, University of Colorado Boulder, and NOAA Chemical Sciences Laboratory, Boulder, CO, USA.

*Correspondence to*: Ewa M Bednarz (ewa.bednarz@noaa.gov)

**Abstract.**

The paper describes the development and performance of the Double Extended Stratospheric-Tropospheric (DEST vn1.0) chemistry scheme, which forms a part of the Met Office's Unified Model coupled to the United Kingdom Chemistry and Aerosol (UM-UKCA) chemistry-climate model, the atmospheric composition model of the United Kingdom Earth System Model (UKESM). The scheme extends the standard Stratospheric-Tropospheric Chemistry scheme (StratTrop) by including a range of important updates to the halogen chemistry. These allow process-oriented studies of stratospheric ozone depletion

and recovery, including the impacts from both controlled long-lived ozone-depleting substances (ODSs) and emerging issues around uncontrolled very short-lived substances (VSLS). The main updates in DEST are (i) an explicit treatment of 14 of the most important long-lived ODSs; (ii) an inclusion of Br-VSLS emissions and chemistry; and (iii) an inclusion of Cl-VSLS emissions/lower boundary conditions and chemistry. We evaluate the scheme's performance by comparing DEST simulations against analogous runs made with the standard StratTrop scheme, as well as against observational and reanalysis datasets.

Overall, our scheme addresses some significant shortcomings in the representation of atmospheric halogens in the standard StratTrop scheme, and will thus be particularly relevant for studies of ozone layer recovery and processes affecting it, in support of future Ozone Assessment reports.

## 1. Introduction

The last two decades have seen extensive international efforts dedicated to the development and application of complex

chemistry-climate models (CCMs) and earth system models (ESMs). By explicitly simulating the interplay of atmospheric chemistry, circulation and the radiative balance of the atmosphere, these models have proved very useful in addressing a range

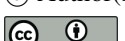



of coupled composition-climate problems of environmental significance, including for example how climate change might impact air quality (e.g. Turnock et al., 2020) and how the stratospheric ozone layer might evolve under a range of potential climate scenarios (e.g. Dhomse et al., 2018). One of the main CCMs developed and used in the United Kingdom is the UK

Met Office's Unified Model coupled to the United Kingdom Chemistry and Aerosol model (UM-UKCA; Morgenstern et al., 2009; O'Connor et al., 2014; Archibald et al., 2020). When coupled to an interactive ocean, sea-ice as well as terrestrial and oceanic biochemistry modules, the model has also been recently known as United Kingdom Earth System Model (UKESM1, Sellar et al, 2019). It currently includes a comprehensive Stratospheric-Tropospheric Chemistry scheme (StratTrop, Archibald et al., 2020) suitable for addressing a wide range of chemistry-climate problems.


Depletion of the stratospheric ozone layer is one of the most prominent environmental issues of the last several decades. While the Montreal Protocol and its amendments are successfully reducing the atmospheric abundance of halogenated long-lived ozone-depleting substances (ODSs), and thus ozone recovery is expected this century (e.g. Dhomse et al., 2018), challenges to our understanding of stratospheric composition, along with potential challenges for the Protocol, have recently emerged.

These include: (1) a persistent downward trend in the extra-polar lower stratospheric ozone (e.g. Ball et al., 2019); (2) illicit production of certain controlled chlorofluorocarbons (CFCs; e.g. Montzka et al., 2018); and (3) increasing emissions of uncontrolled halogenated very short-lived substances (VSLS), such as dichloromethane, $CH_2Cl_2$, and chloroform, $CHCl_3$ (e.g. Fang et al., 2019; Hossaini et al., 2019; Claxton et al., 2020). These issues, among others, are areas of active stratospheric research, relevant to understanding the expected timescale of ozone recovery, along with direct and indirect climate impacts

of ODSs.

The current generation of state-of-the-art CCMs should be equipped to examine topical issues in stratospheric composition research. However, the StratTrop scheme of UKESM1 is not optimized for this purpose, as it does not currently include an explicit representation of most long-lived halogenated ODSs. Instead, total chlorine and bromine contributions from these are

'lumped' into three main 'surrogate' ODS species. Historically, such an approach has been adopted in some CCMs when balancing the need for a reasonable simulation of the stratosphere against the added computational burden of a more complex chemistry scheme that includes a large number of advected tracers, While the lumping approach constitutes a useful approximation of total stratospheric halogen content, it may not produce a fully correct time evolution of halogenated source and product gases that is needed in studies where stratospheric ozone is the primary focus. Notably, an Extended Stratospheric

Chemistry scheme (CheS+; Bednarz et al., 2016), which included an explicit treatment of 12 long-lived ODSs but no tropospheric chemistry, was available in an older UM-UKCA model version (vn7.3) that participated in phase one of the Chemistry-Climate Model Initiative project (CCMI-1, Eyring et al., 2013) in support of the 2018 WMO/UNEP Ozone Assessment Report (WMO, 2018). However, the scheme has not yet been incorporated into the newer UM-UKCA/UKESM versions that include other important improvements. Both the recent CMIP6 (i.e. phase 6 of the Climate Model Intercomparison

Project) UKESM1 simulations (Sellar et al., 2020) and the most recent CCMI-2022 (i.e. phase 2 of CCMI) UKESM1





simulations in support of the 2022 WMO/UNEP Ozone Assessment Report still use the simplified halogen lumping approach of the StratTrop scheme (Archibald et al., 2020).

In addition to longer-lived ODSs, recent studies have showed the importance of chlorinated and brominated very short-lived
substances (Cl-VSLS and Br-VSLS) in contributing to the total halogen budget and, thus, potentially playing an important role in modulating the evolution of stratospheric ozone (e.g. Fernandez et al., 2017; Hossaini et al., 2019). Prominent Cl-VSLS include dichloromethane and chloroform, both of which have significant anthropogenic sources (Chipperfield et al., 2020). Importantly, Cl-VSLS are not yet included in the standard StratTrop scheme of UKESM1. The stratospheric bromine contribution from Br-VSLS, on the other hand, is only roughly approximated in StratTrop by including an extra 5 ppt of
bromine to the lumped lower boundary condition (LBC) of the surrogate gas, methyl bromide ($CH_3Br$). Fernandez et al. (2021) performed a detailed set of sensitivity experiments with the CAM-Chem model to investigate the impact of this surrogate approach versus a fully explicit scheme (i.e. with individual VSLS tracers *emitted* at the surface). Their results highlighted that the latter approach leads to a greater amount of inorganic bromine and ozone destruction in the extra-polar lowermost stratosphere – a region where ozone changes exert relatively large radiative effects (e.g. Hossaini et al., 2015). Wales et al.
(2018) also showed that an explicit treatment of Br-VSLS is important for correctly simulating the distribution of bromine compounds in the lower stratosphere and thus that models relying on the surrogate approach may potentially underestimate bromine-catalysed ozone loss in the region.

Here we describe the development and evaluation of the Double Extended Stratospheric-Tropospheric (DEST v1.0) chemistry
scheme of UM-UKCA (currently at version 11.0). The DEST scheme addresses the above-described shortcomings in the representation of halogen processes in the standard StratTrop scheme of UM-UKCA/UKESM1 (Archibald et al., 2020), including (i) an explicit treatment of 14 most important long-lived ODSs; (ii) an inclusion of Br-VSLS emissions and chemistry; and (iii) an inclusion of Cl-VSLS emissions/LBCs and chemistry. Section 3 provides a detailed account of the chemistry scheme improvements. Section 4.1 describes the experiments performed to evaluate the performance of DEST, and
Sections 4.2-4.3 assess the DEST results against the results of analogous simulations performed with the standard StratTrop scheme and observations/reanalysis. A summary, along with a discussion and outlook is presented in Section 5.

## 2. The UM-UKCA model

We use version 11.0 of the UM-UKCA chemistry-climate model, the atmosphere-only configuration similar to the UKESM1 earth-system model (Sellar, et al., 2019). The full description of the model can be found in Archibald et al. (2020). Briefly, the
model consists of the Global Atmosphere 7.1 configuration of the version 3 of the Hadley Centre Global Environment Model (GA7.1 HadGEM3, Walters et al., 2019) coupled to the UKCA chemistry and aerosol module (Morgenstern et al., 2009). The latter includes the Global Model of Aerosol Processes (GLOMAP-mode) aerosol microphysics module (Mann et al., 2010;

Mulcahy et al., 2018, 2020), as well as the fully interactive FAST-JX photolysis scheme covering the wavelength range 177-850 nm (Telford et al., 2013) up to 0.2 hPa altitude and the look-up tables (Lary and Pyle, 1991) above. The horizontal

resolution of the model used here is 1.875° longitude x 1.75° latitude, with 85 vertical levels up to ~84 km on a terrain-following hybrid height coordinate.

## 3. Chemistry scheme improvements in DEST

The standard UM-UKCA/UKESM1 StratTrop chemistry scheme is described and evaluated in Archibald et al. (2020). The DEST scheme extends the StratTrop (vn11.0) scheme by incorporating important improvements to the representation of

halogen processes. These are described below. A list of all halogenated tracers in DEST is shown in Table 1.

### 3.1. Unlumping of long-lived ODSs

The standard StratTrop scheme explicitly represents only three halogen source gases: $CFCl_3$ (CFC-11), $CF_2Cl_2$ (CFC-12) and $CH_3Br$. In order to reflect changes in total stratospheric halogen content, whose evolution is by and large controlled by the

Montreal Protocol and its subsequent amendments, the imposed LBCs (globally uniform) for these species include halogen contributions from many other major longer-lived ODS. In particular, CFC-11 acts as a surrogate for the chlorine contributions from $CCl_4$, $CH_3CCl_3$, HCFC-141b, HCFC142-b, Halon-1211 and $CH_3Cl$; CFC-12 acts as a surrogate for chlorine from CFC-113, CFC-114, CFC-115, and HCFC-22; and $CH_3Br$ acts as a surrogate for bromine from Halon-1211, Halon-1301, Halon-1202 and Halon-2402. As discussed in the introduction, this 'lumped' halogen approach, while providing a reasonable

approximation of the total stratospheric chlorine and bromine content, is unlikely to accurately represent the evolution of halogenated source and product gases of importance to stratospheric ozone.

To assist future modelling efforts examining, for instance, stratospheric ozone recovery, DEST includes explicit treatment of 14 of the most important ODSs: CFC-11, CFC-12, CFC-113, CFC-114, CFC-115, $CH_3Cl$, $CCl_4$, $CH_3CCl_3$, HCFC-22, HCFC-141b, HCFC-142b, $CH_3Br$, Halon-1211 and Halon-1301 (Table 1). The model time-varying surface concentrations of these

gases are controlled using globally uniform prescribed LBCs. For Halon-1301, the LBCs include also the lumped bromine atom contributions from Halon-1202 and Halon-2402. This approach is similar to the older CheS+ scheme used in Bednarz et al. (2016) but with even more unlumped ODS tracers. In particular, DEST includes CFC-114. CFC-115, HCFC-141b, HCFC-142b as well as lumping of Halon-1202 and Halon-2402 onto Halon-1301, which were not included in the older CheS+ scheme.

### 3.2. Inclusion of Br-VSLS

Wales et al. (2018) and Fernandez et al. (2021) highlighted the importance of including spatially-varying Br-VSLS emissions in models, as opposed to LBCs, for correct representation of tropospheric bromine and its transport to the stratosphere. The



standard StratTrop UM-UKCA scheme does not include explicit representation of Br-VSLS, although their contribution to the total bromine budget is approximated by adding an extra 5 ppt bromine to the LBCs of $CH_3Br$.

In contrast, our new DEST scheme includes explicit treatment of five of the most important naturally-emitted Br-VSLS, namely: bromoform ($CHBr_3$), dibromomethane ($CH_2Br_2$), bromochloromethane ($CH_2BrCl$), dibromochloromethane ($CHBr_2Cl$) and bromodichloromethane ($CHBrCl_2$). The tropospheric abundance of these species now evolves according to prescribed spatially and monthly varying emission fluxes which are based on the climatological emission inventories developed by Ordonez et al. (2012). The annual mean global total emissions for each species are given in Table 2. Note that

these values are slightly smaller than the yearly mean values reported in Ordonez et al. (2012) because of the 360-day calendar used in the default free-running configuration of UM-UKCA. Once emitted, the Br-VSLS tracers can react with OH, O(1D) and Cl (Table 3) as well as undergo photolysis to release inorganic bromine (Table 4).

### 3.3. Inclusion of Cl-VSLS, including COCl$_2$ chemistry

StratTrop does not include any Cl-VSLS chemistry. In contrast, DEST includes explicit treatment of four of the most important Cl-VSLS, namely: dichloromethane ($CH_2Cl_2$), chloroform ($CHCl_3$), perchloroethylene ($C_2Cl_4$) and ethylene dichloride ($C_2H_4Cl_2$). The atmospheric concentrations of these species are constrained at the surface using either LBC or, for $CH_2Cl_2$ and $C_2Cl_4$, spatially varying emission fluxes. The former approach is used in this evaluation (Section 4), as well as in the recent study of Bednarz et al. (2022). The surface lower boundary conditions for each Cl-VSLS are applied in the model in five

latitude bands (90°S-30°S, 30°S-0, 0-30°N, 30°N-60°N and 60°N-90°N) based on the National Oceanic and Atmospheric Administration (NOAA) and Advanced Global Atmospheric Gases Experiment (AGAGE) surface monitoring data and vary annually. Once in the atmosphere, these Cl-VSLS tracers may undergo photolysis (Table 4) and bimolecular reactions with OH and Cl (Table 3). $C_2Cl_4$ is also subject to a termolecular reaction with Cl atoms (Table 5), a reaction shown to improve model measurement agreement of $C_2Cl_4$ profiles in the tropical upper troposphere (Hossaini et al., 2019).


Two organic products following the atmospheric degradation of Cl-VSLS are included, namely a short-lived peroxy species $CHCl_2O_2$ and phosgene ($COCl_2$). $CHCl_2O_2$ is produced from the reaction of $CH_2Cl_2$ with OH or Cl; it can then react with NO, $NO_3$, $HO_2$ or $CH_3O_2$ (Table 3) or undergo photolysis (Table 4). $COCl_2$ is produced from the reactions of $CH_2Cl_2$ and $CHCl_3$ with Cl, as well as from the reaction of $C_2Cl_4$ with OH. These intermediate organic chlorine species, particularly $COCl_2$, are

longer lived than inorganic product gases; this can thus effectively increase the lifetime of atmospheric chlorine and facilitate its transport into the lower stratosphere.

Apart from Cl-VSLS, $COCl_2$ is also produced in the atmosphere from the photolysis of long-lived $CCl_4$ (Table 4) and from the reaction of $CH_3CCl_3$ with OH (Table 3). Assuming this contribution to atmospheric $COCl_2$ from the longer-lived ODSs is





properly accounted for, observed $COCl_2$ can be used to help infer the stratospheric product injection of chlorine resulting from Cl-VSLS emissions (Harrison et al., 2019). In DEST, $COCl_2$ is lost via photolysis, reaction with O(1D) and wet and dry deposition.

### 3.4 Further updates and improvements

A number of other improvements to the halogen chemistry scheme are included in DEST. First, in addition to the new tracers

described in Sections 3.1-3.3, the scheme also includes $Cl_2$, $Br_2$ and $ClNO_2$ as new inorganic halogen tracers (Table 1). These tracers are important products of heterogenous reactions that occur on polar stratospheric clouds (PSCs) and sulfate aerosols (Table 6) but are otherwise not explicitly included in StratTrop. The rates of all bimolecular halogen reactions in DEST have been updated to JPL (2015), and a number of reactions or reaction channels including the newly added $Cl_2$, $Br_2$ and $ClNO_2$ tracers have been added (Tables 3,5,6).


Regarding model deposition processes, in addition to wet deposition of HCl, HOCl, $ClONO_2$, HBr, HOBr and $BrONO_2$ found in standard StratTrop, we have included in DEST also the wet deposition of Cl, ClO, $Cl_2$, $ClNO_2$, Br, $Br_2$ and BrCl using Henry's law constants from Sander (2015), with the exception of ClNO2 which uses the value from Ordonez et al. (2012). Regarding dry deposition, DEST includes dry deposition of $ClONO_2$ and $BrONO_2$ in addition to HCl, HOCl, HBr and HOBr

already included StratTrop.

Regarding photolysis, rather than incorporating the FAST-JX absorption cross-sections used in the older CheS+ scheme (Bednarz et al., 2016), DEST includes updated FAST-JX photolysis cross-sections for CFC-11, CFC-12, CFC-113, $CH_3Cl_3$, HCFC-22, Halon-1211 and Halon-1301 using the SPARC (2013) parametrizations to avoid problems with temperature

dependence of the cross-sections when calculated using the JPL (2015) recommendations (see SPARC, 2013, for details). Regarding the photolysis look-up tables, which are used in the model instead of FAST-JX at the altitudes above 0.2 hPa, the absorption cross-sections for all species were updated following the TOMCAT model (Chipperfield, 1999), with temperature dependence of some of the longer-lived ODSs further modified following SPARC (2013).

Finally, the default DEST scheme includes the extended treatment of heterogenous halogen reactions following Dennison et al. (2019). As detailed in Table 6, in addition to the five standard heterogenous chlorine reactions occurring in StratTrop on Nitric Acid Trihydrate (NAT) PSCs, ice PSCs and sulfate aerosols, the new DEST scheme includes the $ClONO_2$ + HCl reaction on sulfate aerosols, eight new heterogeneous bromine reactions as well as updated uptake coefficients for the heterogeneous chlorine reactions. We note that while these improvements are now available (but optional) in newer UM-UKCA/UKESM1

versions (vn11.3 and later), these were not yet available in the vn11.0 StratTrop version discussed in this paper (Archibald et al., 2020).





## 4. DEST evaluation

### 4.1. Description of the simulations

The performance of our DEST chemistry scheme was evaluated using a 20-year-long (plus spin-up) 'timeslice' UM-UKCA
simulation under perpetual year 2000 conditions. These were created by averaging the forcing data provided for the CMIP6
project (Sellar et al., 2020) over the years 1995-2004 inclusive. These include greenhouse gas and long-lived ODS data from
Meinshausen et al. (2017), sea-surface temperatures and sea-ice from Durack and Taylor (2016), and emissions of aerosols
and chemical tracers of importance in the troposphere as in Archibald et al. (2020) and Sellar et al. (2020).

Lower boundary conditions for Cl-VSLS are representative for the year 2000 and created by averaging surface station data
available in the five latitude bands: 90°S-30°S, 30°S-0°, 0°-30°N, 30°N-60°N and 60°N-90°N. Following Hossaini et al.
(2019), NOAA global monitoring network data were used for $CH_2Cl_2$ and $C_2Cl_4$, and AGAGE network data were used for
$CHCl_3$. The latitude-dependent LBCs for $C_2H_4Cl_2$ were estimated (Hossaini et al., 2016b) based on measurements made during
the 2009-2011 HIPPO aircraft campaign (Wofsy, 2011). The Br-VSLS emissions are also climatological and follow Ordonez
et al. (2012). As a benchmark for comparison with DEST, a second 20-year-long (plus spin-up) simulation was performed
under the same timeslice year 2000 conditions, but which used the standard StratTrop scheme described and evaluated in
Archibald et al. (2020).

In addition to the above timeslice simulations, a 3-member ensemble of an analogous free-running transient (2000-2019)
experiment was performed using the new DEST scheme with time-varying Cl-VSLS LBCs. These transient runs are described
in detail in Bednarz et al., (2022; where they are referred to as 'VSLS'), and here are used for comparison of the model results
with observational and reanalysis datasets.

We note two problems in the code present in the timeslice year 2000 DEST simulations discussed in Section 4.2; note these
are fixed in the final DEST (vn1.0) version lodged in UM-UKCA (vn11.0), as well as in the transient DEST simulations used
here in Section 4.3 and in Bednarz et al. (2022), First, the bimolecular reaction of $COCl_2$ with O(1D) produces one molecule
of CO and ClO each, instead of one molecule of CO, ClO and Cl; this results in a loss of one chlorine atom per reaction.
Second, the update to the photolysis cross-section of CFC-113 after SPARC (2013) discussed in Section 3.4 was not included,
so the reaction proceeds with the old absorption cross-section used in CheS+ scheme in Bednarz et al. (2016). While we
acknowledge that both issues are important and has been fixed in the final DEST vn1.0 that is now lodged in vn11.0 of UM-
UKCA and used for science studies (e.g. Bednarz et al. 2022), the two effects should not have a dominant impact on the
timeslice evaluation results in Section 4.2.





### 4.2. Comparison between DEST and StratTrop

#### 4.2.1. Stratospheric and tropospheric halogens

Figure 1 shows yearly mean differences in total chlorine and bromine volume mixing ratios (i.e. including contributions from both source and product gases) between our new DEST scheme and the standard StratTrop scheme. We find that DEST shows 40-60 ppt more chlorine in the lower stratosphere, with larger differences of up to ~150 ppt simulated in the Northern Hemisphere (NH) troposphere. These differences arise largely from the inclusion of Cl-VSLS in the model, with Cl-VSLS source gases accounting for most of the additional chlorine near the surface (Fig. 2a).


For bromine, we find no substantial differences in total stratospheric bromine content between DEST and StratTrop over large parts of the stratosphere (Fig. 1b). In the lowermost stratosphere, however, DEST simulates higher bromine levels compared to StratTrop (e.g. by 0.3 ppt in the tropics, 25°S-25°N, at 18 km or by 0.2 ppt at 51°N and 15 km). Total bromine levels are also markedly higher in DEST in the tropical and NH troposphere (e.g. up to ~0.8 ppt more bromine in the equatorial upper

troposphere). These increases arise from the inclusion of Br-VSLS in DEST, the elevated levels of which are simulated throughout the troposphere and lowermost stratosphere (Fig. 2b). We calculate the resulting Br-VSLS stratospheric source gas injection in DEST to be 1.9 ppt Br (at 17 km, 20°S-20°N, for the timeslice year 2000 conditions), in a good agreement with the inter-model mean value of 1.5-2.5 ppt derived in Hossaini et al. (2016). We note that in the Southern Hemisphere (SH) troposphere, on the other hand, DEST shows somewhat lower bromine levels compared to StratTrop (up to ~-0.3 ppt in the

midlatitudes).

Regarding long-lived ODSs, DEST shows smaller levels of halogens present in the form of ODS source gases in the lower stratosphere compared to StratTrop, and higher levels in the mid- and upper stratosphere (Fig. 3). For bromine, we also find 4-5 ppt less bromine in long-lived ODSs in the troposphere (Fig. 3b); this in accord with the 5 ppt bromine included in StratTrop

in the LBC of $CH_3Br$ to account for the bromine contribution from Br-VSLS (Section 3.2), which in DEST is instead represented explicitly (Fig. 2b).

Figure 4 shows the associated differences in in some of the main inorganic halogen species. The elevated tropospheric and lower stratospheric chlorine levels in DEST (Fig. 1a) compared to StratTrop increase HCl and $ClONO_2$ – the main chlorine

reservoirs – in the lower stratosphere (Fig. 4a-b). Above, in the mid- and upper stratosphere, DEST shows a decrease in HCl and $ClONO_2$, in accord with the concurrent higher levels of chlorine present in the form of long-lived ODSs (Fig. 3a).

For bromine, DEST shows an increase in lower stratospheric BrO and a decrease in BrO above (Fig. 4f); this is thus qualitatively similar to what was found for HCl. The BrO increase is related to the higher total bromine levels simulated in the

lowermost stratosphere as the result of including Br-VSLS (as discussed above) as well as to the inclusion of heterogenous

bromine reactions (Section 3.4). The decrease in BrO directly above arises because of concurrent increase in bromine levels found in the from of long-lived ODSs at these altitudes (Fig. 3b). For $BrONO_2$, its concentrations are lower in DEST throughout the stratosphere (Fig. 4e) compared to StratTrop, and this is partially related to the concurrent increase in stratospheric concentrations of BrCl (Fig. 4c). Regarding HBr, DEST shows significant increase in HBr throughout the troposphere as the
result of the increase in total tropospheric bromine from including Br-VSLS, with a decrease in HBr simulated in the stratosphere (Fig. 4d).

Figure 3c also shows the corresponding annual mean difference in total reactive chlorine between DEST and StratTrop. We find markedly higher levels of reactive chlorine in DEST in the high latitude lower stratosphere in both hemispheres, i.e. by
up to ~140 ppt in the SH and ~60 ppt in the NH in yearly mean. These yearly mean values correspond to accelerated heterogenous reactions on PSCs and aerosols inside the polar vortices in winter and spring. The response reflects the combined impact of the updates to the heterogeneous halogen reactions (Section 3.4), as already discussed in Dennison et al. (2019), and the increase in total stratospheric chlorine in DEST compared to StratTrop (Fig. 1a) as the result of including Cl-VSLS.

### 4.2.2. Stratospheric ozone

The increase in reactive chlorine (Fig. 4) and bromine (BrO and BrCl, Fig. 3f and 3c) in the lower stratosphere in DEST has important consequences for stratospheric ozone concentrations simulated in the model (Fig. 5). We find significant reductions in lower stratospheric ozone levels throughout the globe in DEST compared to StratTrop, with up to ~4% decrease in lower stratospheric ozone simulated in the tropics (Fig. 5a). In the high latitudes during spring, the ozone reduction reaches locally ~75% in the SH (Fig. 5b) and ~20% in the NH (Fig. 5c). When integrated over the depth of the atmosphere, statistically
significant decreases in total column $O_3$ are generally found in DEST in the mid-latitudes, with up to 16 DU and 20 DU lower total column $O_3$ found at 60°S and 60°N in October and March, respectively (Figure 5d).

### 4.2.3. Stratospheric climate

The reduction in stratospheric ozone levels in DEST relative to StratTrop affects stratospheric climate. The decrease in ozone in the tropical lower stratosphere results in a small but statistically significant cooling of ~0.2 K in the region (Fig. 6a). The
decrease in tropical cold point temperatures in turn reduces the amount of water vapour entering the stratosphere, resulting in up ~3% lower stratospheric water vapour levels in DEST compared to StratTrop (Fig. 6b). Changes in lower stratospheric temperatures also impact the large-scale transport, manifested by the slightly younger stratospheric age-of-air (AoA) (Fig. 6c) presumably as the consequence of reduction in tropospheric static stability. The changes in transport further impact the concentrations of halogenated source and product gases (Section 4.2.1).


In the SH high latitudes, the DEST simulation also shows a local cooling in the lower stratosphere and a warming above. The response is a signature of an accelerated Antarctic springtime ozone depletion and its impact on the SH polar vortex, as was



found in a number of studies in the context of the impact of ODSs on Antarctic ozone (e.g. MacLandress et al., 2011; Keeble et al., 2014).

**4.3. Comparison between DEST and observations/reanalysis**

We now evaluate the performance of DEST against observations and reanalysis. In order to facilitate better comparison with observational datasets, rather than the timeslice year 2000 simulation discussed in Section 4.2., we use the three-member ensemble of transient 2000-2019 integrations described in Bednarz et al. (2022). In addition to using time-varying forcings, these use the final DEST version where the two problems in the code present in the timeslice year 2000 integrations discussed in Section 4.2. have been fixed (see Section 4.1).

Figures 7 and 8 compare the surface concentrations of $CHBr_3$ and $CH_2Br_2$ measured over 2010-2018 at a number of NOAA monitoring sites (updated from Hossaini et al., 2016a; see also https://gml.noaa.gov, last accessed 29 August 2022) to those simulated in DEST at the same locations. We find that the inclusion of Br-VSLS emissions result in relatively good agreement with observations at most of the sites analysed. Some exceptions remain, however, especially for bromoform, whose simulated concentrations tend to be too small in the NH mid- and high latitudes and too large at some of the sites in the NH subtropics.

Figure 9 compares the DEST-simulated HCl, ozone, $COCl_2$ and water vapour levels with the ACE-FTS (vn3.5-3.6) satellite data (Boone et al., 2013); and Figure 10 compares the DEST simulated temperatures and zonal winds to the ERA-Interim reanalysis (Dee et al., 2011).We find that the DEST runs overestimate HCl in the tropical (up to ~0.2 ppb) and high-latitude (up to ~0.8 ppb over the Antarctic) lower stratosphere, and underestimate HCl over the rest of the stratosphere (by up to ~-0.2 ppb), compared to ACE-FTS (Fig. 9a). Bednarz et al. (2022) showed that the inclusion of Cl-VSLS in the model increases HCl throughout the stratosphere; this effect thus acts to improve the model-measurement comparison in the mid- and upper stratosphere.

For ozone, DEST shows too high ozone concentrations in the tropical upper troposphere and lower stratosphere (Fig. 9b). The reduction in tropical lower stratospheric ozone in DEST compared to StratTrop (Fig. 5a) implies that DEST performs better in this respect than StratTrop. The opposite is true for the high-latitudes: the transient DEST runs underestimate lower stratospheric ozone in the polar regions compared to ACE-FTS (Fig. 9b), and the reduction in polar ozone in DEST compared to StratTrop (Fig. 5a) worsens the comparison with satellite data. We note that the comparison is made only with one satellite dataset, while important uncertainties exist in most observational datasets. For instance, a small positive bias of a few percent in the lower stratospheric ozone concentrations was reported for vn3.6 of ACE-FTS data used here (Sheese et al., 2022).

For phosgene (Fig. 9c), DEST compares reasonably well in the tropics at ~20 km altitude, but underestimate phosgene levels

in other regions. $COCl_2$ is not included at all in StratTrop, and so it follows that the standard scheme omits this important stratospheric chlorine species.

Regarding stratospheric water vapour, DEST overestimates stratospheric water vapour compared to ACE-FTS (up to ~2.0-2.5 ppm in the tropical lower stratosphere; Fig. 9d); this likely arises because of markedly warmer tropical lower stratosphere in

the model (by ~2 K at 100 hPa compared to ERAI-Interim, Fig. 10a) facilitating too much input of water vapour to the stratosphere. Since the reduction in tropical lower stratospheric ozone in DEST compared to StratTrop (Fig. 5a) was shown to reduce tropical cold point temperatures (Fig. 6a) and thus stratospheric water vapour levels (Fig. 6b), it follows that DEST compares better to ACE-FTS in this respect than the standard StratTrop version.

Finally, in the SH high latitudes DEST simulates a too cold lower stratosphere and too strong SH polar vortex compared to ERAI-Interim (Fig. 8). Since the reduction in Antarctic ozone in DEST compared to StratTrop results in cooling in the Antarctic lower stratosphere (Fig. 6a), it follows that in this respect the comparison with reanalysis is worse in DEST than for the standard StratTrop scheme.

To summarize, we find the impact of the new DEST developments (compared to StratTrop) improves some aspects of the comparison with satellite observations and reanalysis, although it also worsens other aspects. We note, however, that the simulated concentrations of atmospheric tracers and fields are the cumulative result of a range of chemical, radiative and dynamical processes, and so an improvement in a small subset of these does not necessarily guarantee a better model agreement with observations/reanalysis. In addition, the study uses only one satellite dataset (ACE-FTS vn3.5-3.6) and one reanalysis

(ERA-Interim), while substantial uncertainties exist in most of the observational and reanalysis datasets, and differences are commonly found between different satellite or reanalysis products (e.g. SPARC, 2018).

## 5. Summary and outlook

We have described the development and performance of the new Double Extended Stratospheric-Tropospheric (DEST vn1.0) chemistry scheme, which constitutes a part of the UM-UKCA chemistry-climate model, the atmospheric composition model

of UKESM1 (Sellar et al., 2019). The DEST scheme is an extension of the standard Stratospheric-Tropospheric Chemistry scheme (StratTrop; Archibald et al., 2020) that includes a range of important updates to the halogen chemistry allowing process-oriented studies of stratospheric ozone depletion and recovery, including the impacts from both controlled long-lived ozone depleting substances and uncontrolled halogenated very-short substances. The main updates in DEST are (i) an explicit treatment of 14 of the most important long-lived ODSs; (ii) an inclusion of Br-VSLS emissions and chemistry; and (iii) an

inclusion of Cl-VSLS emissions/LBCs and chemistry. Further updates include the inclusion of additional inorganic halogen

tracers and changes to the photolysis, gas-phase and heterogeneous reaction rates (the latter following Dennison et al., 2019). As a result, the DEST scheme improves on some of the important shortcomings in the representation of halogens in the standard StratTrop scheme that is currently being used in the UKESM1 CCMI-2022 simulations in support of the UNEP/WMO Ozone Assessment Reports and, thus, will be particularly relevant for studies informing the future reports.


The performance of DEST was evaluated against the standard StratTrop scheme as well as against the ACE-FTS satellite observations and the ERA-Interim reanalysis. We found larger lower stratospheric total chlorine levels (~40-60 ppt Cl for the year 2000 conditions) in DEST compared to StratTrop, as well as significant changes in tropospheric total chlorine and bromine levels and their horizontal distributions. Total stratospheric bromine levels were found to be similar between the DEST and

StratTrop over large parts of the stratosphere, with the exception of the lowermost stratosphere where DEST showed higher bromine levels (e.g. by 0.3 ppt in the tropics at 18 km altitude). The changes in total halogen levels in the stratosphere were accompanied by marked changes in the speciation of inorganic halogen species as well as in the levels of halogens found in the form of longer-lived ODSs. The resulting impacts on stratospheric ozone, water vapour, temperature and transport were also discussed.


Future improvements in DEST vn2.0 will incorporate iodine chemistry, which has now emerged as a potentially important contributor to both processes in the troposphere as well as to stratospheric ozone depletion (e.g. Koenig et al., 2020; Cuevas et al., 2022).

**Acknowledgements**

EMB was supported by the UK Natural Environment Research Council (NERC) SISLAC project (NE/R001782/1). RH was supported by the NERC Independent Research Fellowship (NE/N014375/1), the NERC ISHOC project (NE/R004927/1), and the NERC SISLAC project (NE/R001782/1). MPC was supported by the NERC NCEO project UKESM and TerraFirma.

The simulations were carried out using MONSooN2, a collaborative High-Performance Computing facility funded by the Met

Office and the NERC, and using the ARCHER UK National Supercomputing Service. The UM and/or JULES code branch(es) used in the publication have not all been submitted for review and inclusion in the UM/JULES trunk or released for general use.

The authors thank Steve Montzka for providing the NOAA $CHBr_3$ and $CH_2Br_2$ data, and Mohit Dalvi for setting up the

StratTrop timeslice simulation.

**Data and code availability**

The data from all UM-UKCA simulations used in this manuscript, as well as plotting scripts used to make all figures, are available from https://doi.org/10.5281/zenodo.7033255 (Bednarz, 2022).. ACE-FTS data can be obtained from http://www.ace.uwaterloo.ca/data.php (last accessed 18 July 2022). ERA-Interim data can be obtained from
https://www.ecmwf.int/en/forecasts/datasets/reanalysis-datasets/era-interim (last accessed 18 July 2022). The NOAA surface measurements of $CHBr_3$ and $CH_2Br_2$ will be available from https://gml.noaa.gov/; please contact Steve Montzka if you require access to it earlier.

All simulations used in this work were performed using version 11.0 of the Met Office Unified Model coupled to the United
Kingdom Chemistry and Aerosol model (UM-UKCA). The UM and/or JULES code branch(es) used in the publication have not all been submitted for review and inclusion in the UM/JULES trunk or released for general use. However, the UM and JULES code branches were made available to reviewers of this manuscript. Due to intellectual property right restrictions, we cannot provide the source code for UM-UKCA. The UM-UKCA model is available for use through a licensing agreement. A number of research organisations and national meteorological services use UM-UKCA in collaboration with the Met Office
to undertake basic atmospheric process research, produce forecasts, develop the model code and build and evaluate earth system models. Please visit https://www.metoffice.gov.uk/research/approach/modelling-systems/unified-model (last accessed 15 July 2022) for further information on how to apply for a licence.

**Authors contributions**

EMB performed the UM-UKCA chemistry scheme developments, with technical guidance from NLA and scientific guidance
from RH. EMB performed the DEST simulations, analysed the results and wrote the first draft of the manuscript. All authors contributed to the discussion of the results and writing of the manuscript.

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




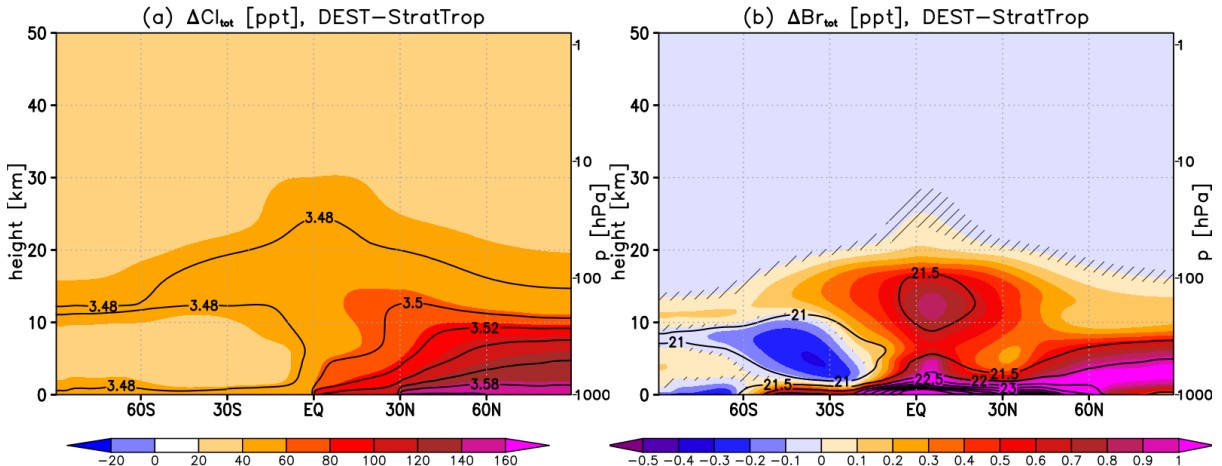

**Figure 1.** Shading: Annual mean difference [ppt] in simulated total Cl (a) and total Br (b) between DEST and StratTrop for the year 2000 conditions. Hatching denotes regions where the difference is not statistically significant, here taken as being lower than ±2 standard errors. Contours show the corresponding values in DEST for reference; note that in panel (a) this is plotted in units of ppb.


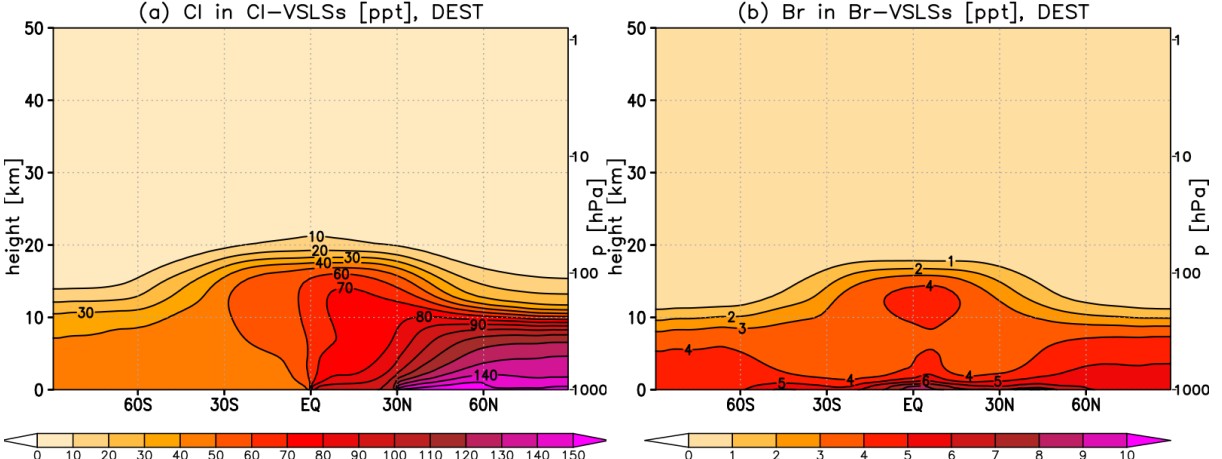

**Figure 2.** Annual mean Cl in Cl-VSLSs (a) and Br in Br-VSLSs (b) [ppt] simulated in DEST for the year 2000 conditions.



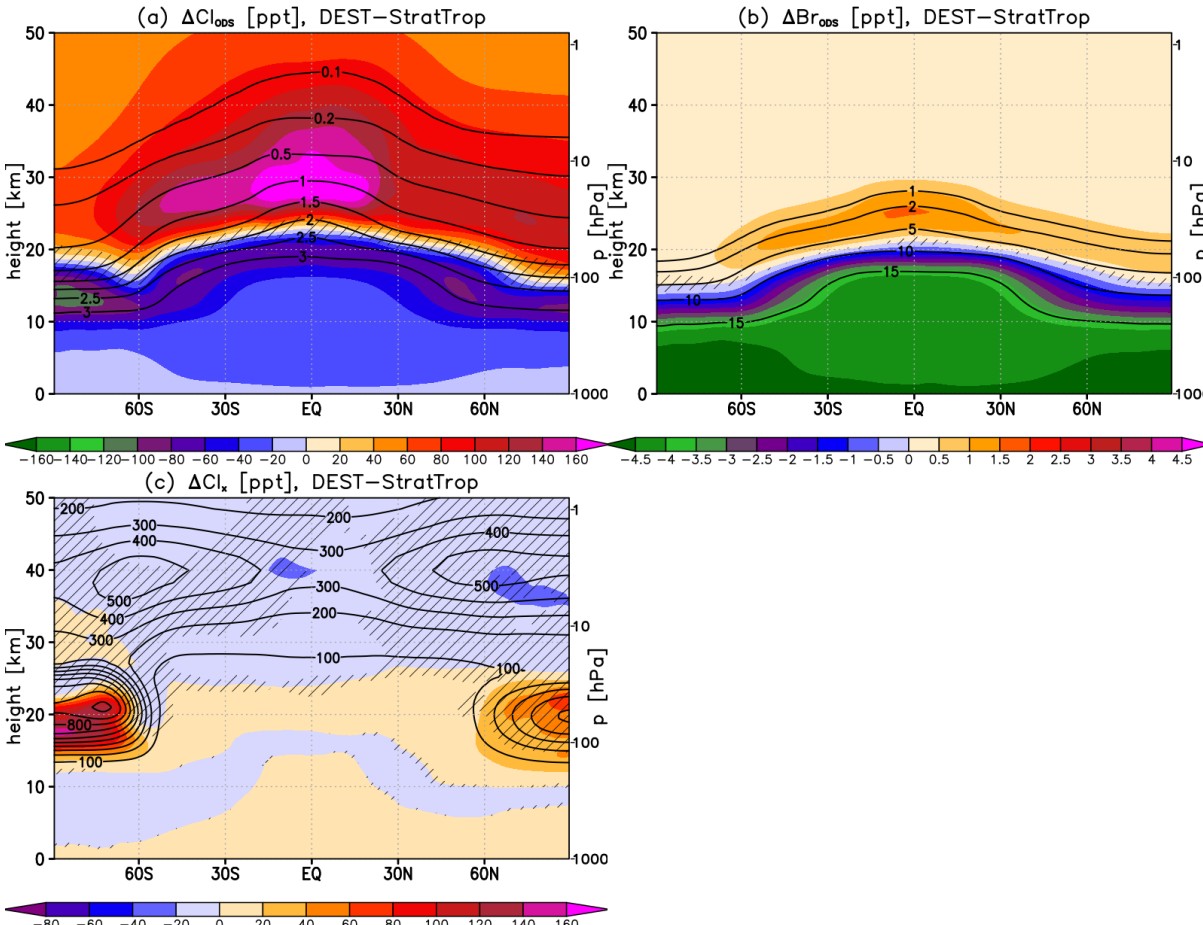

**Figure 3. Shading: Annual mean difference [ppt] in Cl present in long-lived ODSs, $Cl_{ODS}$ (a), Br present in long-lived ODSs, $Br_{ODS}$ (b), and total reactive chlorine, $Cl_x$ (with $Cl_x = ClO + 2 \cdot Cl_2O_2 + Cl + OClO + 2 \cdot Cl_2$) between DEST and StratTrop for the year 2000 conditions. Hatching denotes regions where the difference is not statistically significant, here taken as being lower than ±2 standard errors. Contours show the corresponding values in DEST for reference; note that in panel (a) this is plotted in units of ppb.**






**Figure 4.** Shading: Annual mean difference [ppt] in simulated HCl (a), ClONO₂ (b), BrCl (c), HBr (d), BrONO₂ (e) and BrO (f) between DEST and StratTrop for the year 2000 conditions. Hatching as in Fig. 1. Contours show the corresponding values in DEST for reference; note that in panels (a) and (b) this is plotted in units of ppb.


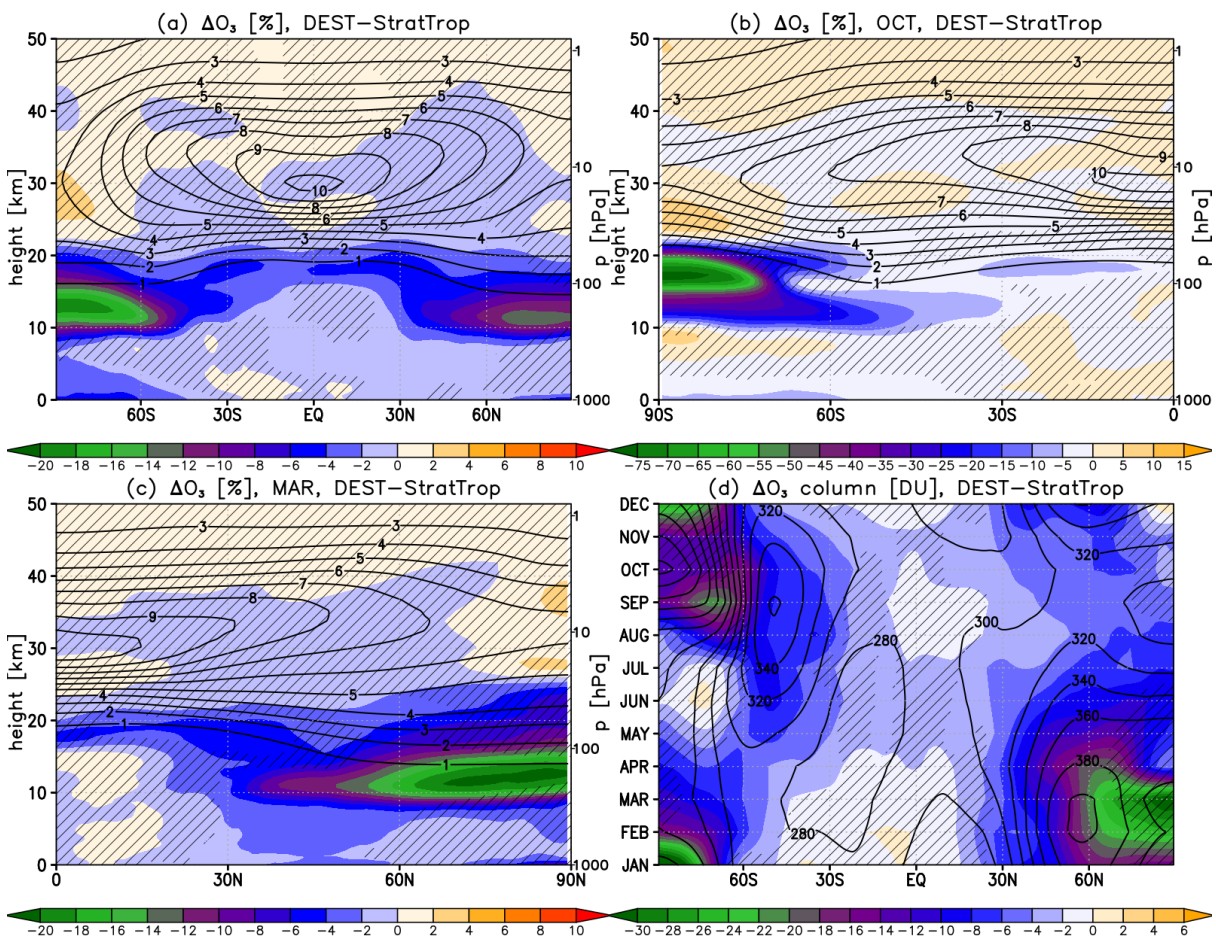

**Figure 5.** Shading: The difference in annual mean $O_3$ [%] (a), Southern Hemispheric $O_3$ [%] in October (b), Northern Hemispheric $O_3$ [%] in March (c), and total column $O_3$ as a function of latitude and month [DU](d) between DEST and StratTrop for the year 2000 conditions. Hatching as in Fig. 1. Contours show the corresponding values in DEST for reference (in units of ppm for (a-c), and in units of DU in (d)).



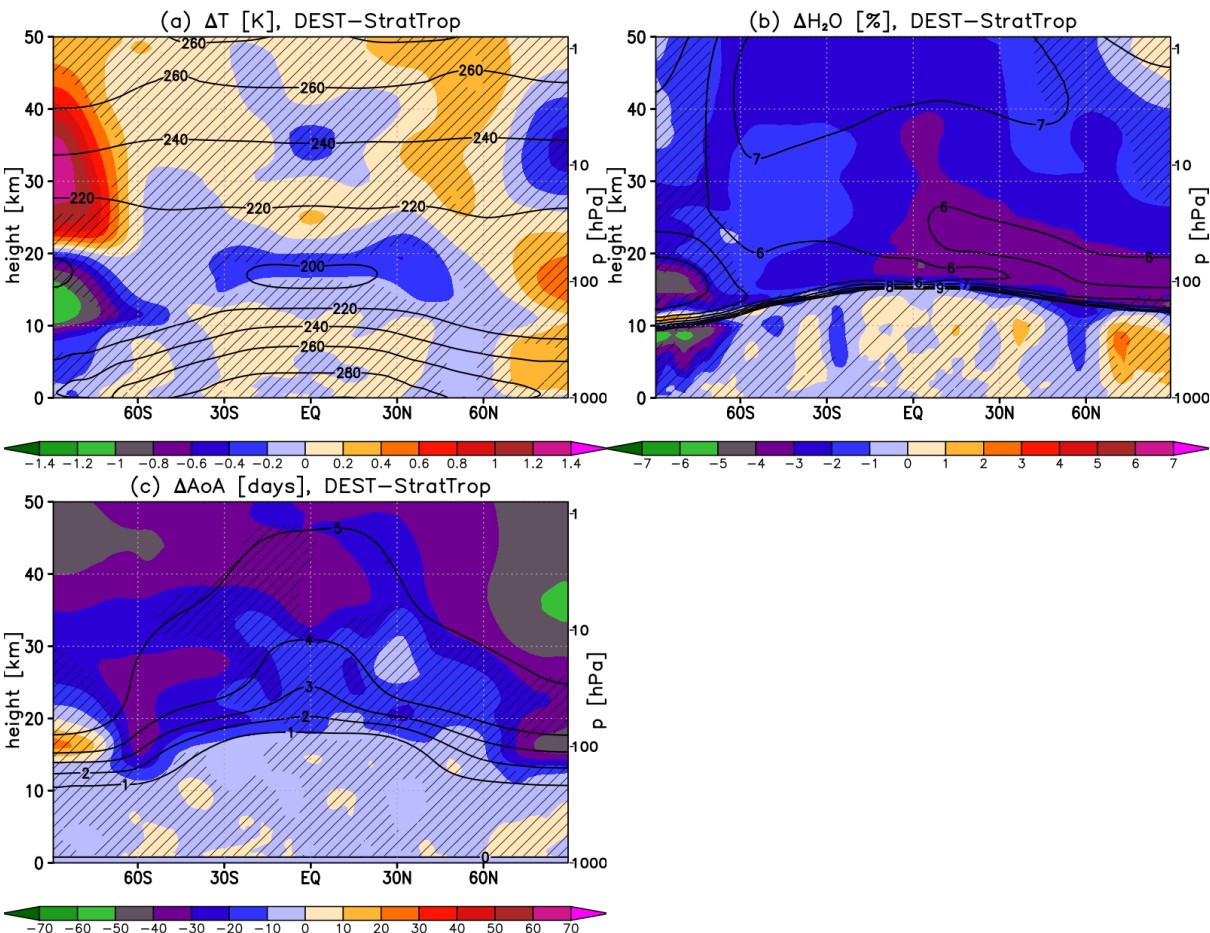

**Figure 6. As in Fig. 4 but for the difference in temperature [K] (a), specific humidity [%] (b) and model age of air [days] (c). Note contours in panels (b) and (c) are in the units of ppm and years, respectively.**







**Figure 7. Monthly mean evolution of 2010-2018 climatological surface CHBr₃ concentrations [ppt] measured by different NOAA measurement sites (updated from Hossaini et al., 2016a; black), and the corresponding concentrations simulated at these locations in the individual ensemble members of the transient DEST experiment (red). The code names for the different sites are: SPO – South Pole, Antarctica; CGO- Cape Grim, Australia; SMO - Tutuila, American Samoa; MLO - Mauna Loa, HI, USA; KUM - Cape Kumukai, HI, USA; NWR - Niwot Ridge, CO, USA; BRW - Barrow, AK, USA; ALT - Alert, Canada; PSA - Palmer Station, Antarctica; LEF - Park Falls, WI, USA; HFM - Harvard Forest, MA, USA; MHD -Mace Head, Ireland; THD - Trinidad Head, CA, USA. The DEST values at THD reach 8.6-9.9 ppt in October for the individual ensemble members.**








**Figure 8. As in Fig. 7 but for CH₂Br₂ [ppt].**





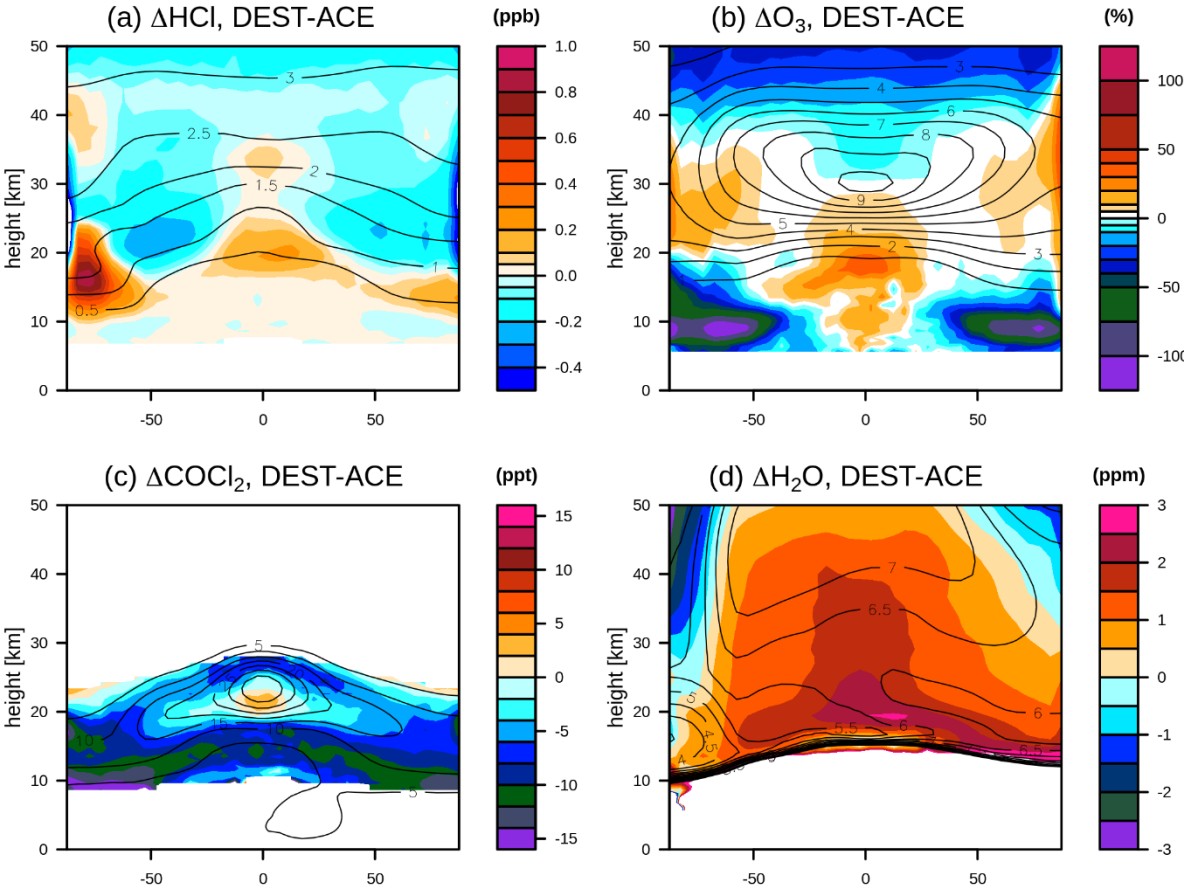

**Figure 9.** Shading: 2005-2019 annual mean difference in (a) HCl [ppb], (b) $O_3$ [%], (c) $COCl_2$ [ppt] and (d) H2O [ppm] between the ensemble mean transient DEST simulations and ACE-FTS vn3.5-3.6 data. Contours show the corresponding DEST climatology for reference. The percentage difference in (b) is calculated relative to the model values.



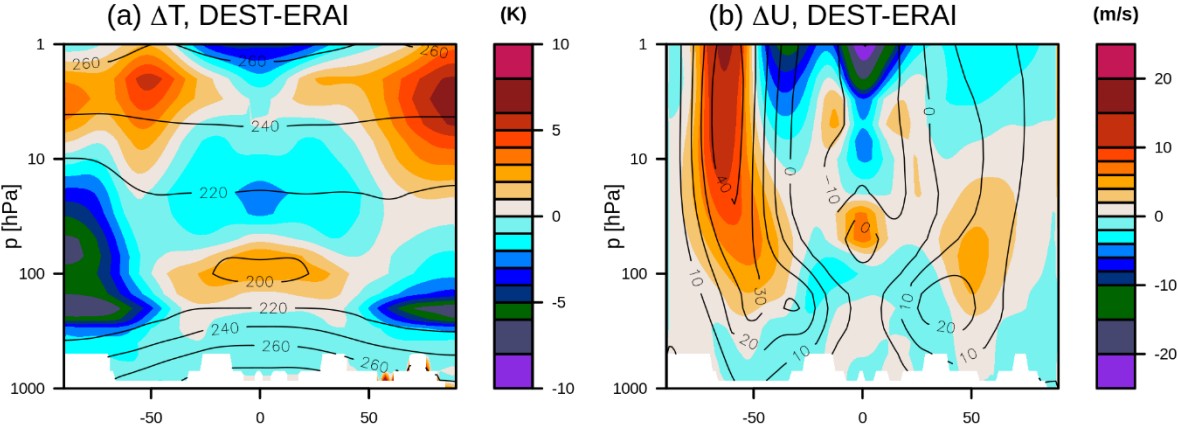

**Figure 10. Shading: 2005-2018 annual mean difference in zonal mean (a) temperature [K] and (b) zonal wind [m/s] between the ensemble mean transient DEST simulations and ERA-Interim reanalysis. Contours show the corresponding DEST climatology for reference.**





| Species type | Species formula |
|---|---|
| Cl$_y$ | Cl, **Cl$_2$**, Cl$_2$O$_2$, ClO, OClO, HCl, HOCl, ClONO$_2$, **ClNO$_2$** |
| Br$_y$ | Br, **Br$_2$**, BrO, BrONO$_2$, HBr, HOBr |
| mixed Cl$_y$/Br$_y$ | BrCl |
| long-lived ODSs | CFCl$_3$ (CFC11), CF$_2$Cl$_2$ (CFC12), **CF$_2$ClCFCl$_2$ (CFC113)**, **CF$_2$ClCF$_2$Cl (CFC114)**, **CF$_2$ClCF$_3$ (CFC115)**, **CH$_3$Cl**, **CCl$_4$**, **CH$_3$CCl$_3$**, **CHF$_2$Cl (HCFC22)**, **CH$_3$CFCl$_2$ (HCFC141b)**, **CH$_3$CF$_2$Cl (HCFC142b)**, CH$_3$Br, **CF$_2$ClBr (Halon1211)**, **CF$_3$Br (Halon1301)** |
| Br-VSLSs | **CHBr$_3$**, **CH$_2$Br$_2$**, **CH$_2$BrCl**, **CHBr$_2$Cl**, **CHBrCl$_2$** |
| Cl-VSLSs | **CH$_2$Cl$_2$**, **CHCl$_3$**, **C$_2$Cl$_4$**, **C$_2$H$_4$Cl$_2$** |
| organic mid-products | **COCl$_2$**, **CHCl$_2$O$_2$** |

**Table 1. Summary of the halogen tracers included in DEST. New species added in DEST that are absent in the standard StratTrop scheme are highlighted in bold font and underlined. Note that in StratTrop, CFC11, CFC12 and CH$_3$Br constitute surrogate species reflecting lumped halogen contributions from other long-lived ODSs, see Section 3.1 for details.**

| Species | Total emissions [Gg/yr] |
|---|---|
| CHBr$_3$ | 528 |
| CH$_2$Br$_2$ | 67 |
| CH$_2$BrCl | 10 |
| CHBr$_2$Cl | 20 |
| CHBrCl$_2$ | 22 |

**Table 2. Annual mean global total emissions of Br-VSLS imposed in the model based on the inventory of Ordonez et al. (2012).**






| Reactants | Products | Rate constant | Source |
|---|---|---|---|
| Br + Cl$_2$O$_2$ | → BrCl + Cl + O$_2$ | $5.90 \cdot 10^{-12} \cdot \exp(-170/T)$ | JPL2015 |
| Br + HCHO | → HBr + CO + HO$_2$ | $1.70 \cdot 10^{-11} \cdot \exp(-800/T)$ | JPL2015 |
| Br + HO$_2$ | → HBr + O$_2$ | $4.80 \cdot 10^{-12} \cdot \exp(-310/T)$ | JPL2015 |
| Br + O$_3$ | → BrO + O$_2$ | $1.60 \cdot 10^{-11} \cdot \exp(-780/T)$ | JPL2015 |
| Br + OClO | → BrO + ClO | $2.60 \cdot 10^{-11} \cdot \exp(-1300/T)$ | JPL2015 |
| BrO + BrO | → Br + Br + O$_2$ | $2.40 \cdot 10^{-12} \cdot \exp(40/T)$ | JPL2015 |
|  | → **Br$_2$ + O$_2$** | $2.80 \cdot 10^{-14} \cdot \exp(860/T)$ | JPL2015 |
| BrO + ClO | → Br + Cl + O$_2$ | $2.30 \cdot 10^{-12} \cdot \exp(260/T)$ | JPL2015 |
|  | → Br + OClO | $9.50 \cdot 10^{-13} \cdot \exp(550/T)$ | JPL2015 |
|  | → BrCl + O$_2$ | $4.10 \cdot 10^{-13} \cdot \exp(290/T)$ | JPL2015 |
| BrO + HO$_2$ | → HOBr + O$_2$ | $4.50 \cdot 10^{-12} \cdot \exp(460/T)$ | JPL2015 |
| BrO + NO | → Br + NO$_2$ | $8.80 \cdot 10^{-12} \cdot \exp(260/T)$ | JPL2015 |
| BrO + OH | → Br + HO$_2$ | $1.70 \cdot 10^{-11} \cdot \exp(250/T)$ | JPL2015 |
| CF$_2$Cl$_2$ + O($^1$D) | → Cl + ClO | $1.20 \cdot 10^{-10} \cdot \exp(25/T)$ | JPL2015 |
| CFCl$_3$ + O($^1$D) | → Cl + Cl + ClO | $2.07 \cdot 10^{-10}$ | JPL2015 |
| Cl + CH$_4$ | → HCl + CH$_3$O$_2$ | $7.10 \cdot 10^{-12} \cdot \exp(-1270/T)$ | JPL2015 |
| Cl + Cl$_2$O$_2$ | → 3·Cl | $7.60 \cdot 10^{-11} \cdot \exp(65/T)$ | JPL2015 |
| Cl + ClONO$_2$ | → 2·Cl + NO$_3$ | $6.50 \cdot 10^{-12} \cdot \exp(135/T)$ | JPL2015 |
| Cl + H$_2$ | → HCl + H | $3.05 \cdot 10^{-11} \cdot \exp(-2270/T)$ | JPL2015 |
| Cl + H$_2$O$_2$ | → HCl + HO$_2$ | $1.10 \cdot 10^{-11} \cdot \exp(-980/T)$ | JPL2015 |
| Cl + HCHO | → HCl + CO + HO$_2$ | $8.10 \cdot 10^{-11} \cdot \exp(-30/T)$ | JPL2015 |
| Cl + HO$_2$ | → ClO + OH | $3.60 \cdot 10^{-11} \cdot \exp(-375/T)$ | JPL2015 |
|  | → HCl + O$_2$ | $1.40 \cdot 10^{-11} \cdot \exp(270/T)$ | JPL2015 |





| | | | |
|---|---|---|---|
| Cl + HOCl | → 2·Cl + OH | $3.40 \cdot 10^{-12} \cdot \exp(-130/T)$ | JPL2015 |
| Cl + MeOOH | → HCl + $CH_3O_2$ | $5.70 \cdot 10^{-11}$ | JPL2015 |
| Cl + $NO_3$ | → ClO + $NO_2$ | $2.40 \cdot 10^{-11}$ | JPL2015 |
| Cl + $O_3$ | → ClO + $O_2$ | $2.30 \cdot 10^{-11} \cdot \exp(-200/T)$ | JPL2015 |
| Cl + OClO | → 2· ClO | $3.40 \cdot 10^{-11} \cdot \exp(160/T)$ | JPL2015 |
| ClO + ClO | **→ $Cl_2$ + $O_2$** | $1.00 \cdot 10^{-12} \cdot \exp(-1590/T)$ | JPL2015 |
| | **→ $Cl_2$ + $O_2$** | $3.00 \cdot 10^{-11} \cdot \exp(-2450/T)$ | JPL2015 |
| | → Cl + OClO | $3.50 \cdot 10^{-13} \cdot \exp(-1370/T)$ | JPL2015 |
| ClO + $HO_2$ | → HOCl + $O_2$ | $2.60 \cdot 10^{-12} \cdot \exp(290/T)$ | JPL2015 |
| ClO + MeOO | → Cl + HCHO + $HO_2$ | $1.80 \cdot 10^{-12} \cdot \exp(-600/T)$ | JPL2015 |
| ClO + NO | → Cl + $NO_2$ | $6.40 \cdot 10^{-12} \cdot \exp(290/T)$ | JPL2015 |
| ClO + $NO_3$ | → Cl + $O_2$ + $NO_2$ | $4.70 \cdot 10^{-13}$ | JPL2015 |
| $CH_3Br$ + Cl | → Br + HCl | $1.46 \cdot 10^{-11} \cdot \exp(-1040/T)$ | JPL2015 |
| $CH_3Br$ + $O(^1D)$ | → Br + OH | $1.80 \cdot 10^{-10}$ | JPL2015 |
| $CH_3Br$ + OH | → Br + $H_2O$ | $1.42 \cdot 10^{-12} \cdot \exp(-1150/T)$ | JPL2015 |
| Br + $NO_3$ | → BrO + $NO_2$ | $1.60 \cdot 10^{-11}$ | JPL2015 |
| HBr + $O(^1D)$ | → HBr + $O(^3P)$ | $3.00 \cdot 10^{-11}$ | JPL2015 |
| | → OH + Br | $9.00 \cdot 10^{-11}$ | |
| | **→ H + BrO** | $3.00 \cdot 10^{-11}$ | |
| HCl + $O(^1D)$ | → H + ClO | $3.30 \cdot 10^{-11}$ | JPL2015 |
| | → $O(^3P)$ + HCl | $1.80 \cdot 10^{-11}$ | |
| | → OH + Cl | $9.90 \cdot 10^{-11}$ | |
| BrO + $O(^3P)$ | → $O_2$ + Br | $1.90 \cdot 10^{-11} \cdot \exp(230/T)$ | JPL2015 |
| ClO + $O(^3P)$ | → Cl + $O_2$ | $2.80 \cdot 10^{-11} \cdot \exp(85/T)$ | JPL2015 |





| | | | |
|---|---|---|---|
| $ClONO_2 + O(^3P)$ | $\rightarrow ClO + NO_3$ | $3.60 \cdot 10^{-12} \cdot \exp(-840/T)$ | JPL2015 |
| $HBr + O(^3P)$ | $\rightarrow OH + Br$ | $5.80 \cdot 10^{-12} \cdot \exp(-1500/T)$ | JPL2015 |
| $HCl + O(^3P)$ | $\rightarrow OH + Cl$ | $1.00 \cdot 10^{-11} \cdot \exp(-3300/T)$ | JPL2015 |
| $HOCl + O(^3P)$ | $\rightarrow OH + ClO$ | $1.70 \cdot 10^{-13}$ | JPL2015 |
| $OClO + O(^3P)$ | $\rightarrow O_2 + ClO$ | $2.40 \cdot 10^{-12} \cdot \exp(-960/T)$ | JPL2015 |
| $ClO + OH$ | $\rightarrow HCl + O_2$ | $6.00 \cdot 10^{-13} \cdot \exp(230/T)$ | JPL2015 |
| | $\rightarrow HO_2 + Cl$ | $7.40 \cdot 10^{-12} \cdot \exp(270/T)$ | JPL2015 |
| $ClONO_2 + OH$ | $\rightarrow HOCl + NO_3$ | $1.20 \cdot 10^{-12} \cdot \exp(-330/T)$ | JPL2015 |
| $HCl + OH$ | $\rightarrow H_2O + Cl$ | $1.80 \cdot 10^{-12} \cdot \exp(-250/T)$ | JPL2015 |
| $HOCl + OH$ | $\rightarrow ClO + H_2O$ | $3.00 \cdot 10^{-12} \cdot \exp(-500/T)$ | JPL2015 |
| $OClO + OH$ | $\rightarrow HOCl + O_2$ | $1.40 \cdot 10^{-12} \cdot \exp(600/T)$ | JPL2015 |
| **$CFCl_3 + OH$** | **$\rightarrow 2 \cdot Cl + ClO$** | $1.00 \cdot 10^{-11} \cdot \exp(-9700/T)$ | JPL2015 |
| **$CF_2Cl_2 + OH$** | **$\rightarrow Cl + ClO$** | $1.00 \cdot 10^{-11} \cdot \exp(-11900/T)$ | JPL2015 |
| **$CH_3Cl + OH$** | **$\rightarrow Cl + H_2O$** | $1.96 \cdot 10^{-12} \cdot \exp(-1200/T)$ | JPL2015 |
| **$CH_3Cl + O(^1D)$** | **$\rightarrow ClO$** | $2.34 \cdot 10^{-10}$ | JPL2015 |
| **$CF_2ClBr + OH$** | **$\rightarrow Cl + BrO$** | $1.00 \cdot 10^{-12} \cdot \exp(-3500/T)$ | JPL2015 |
| **$CF_2ClBr + O(^1D)$** | **$\rightarrow Cl + BrO$** | $9.75 \cdot 10^{-11}$ | JPL2015 |
| **$CCl_4 + OH$** | **$\rightarrow 3 \cdot Cl + ClO$** | $1.00 \cdot 10^{-11} \cdot \exp(-6200/T)$ | JPL2015 |
| **$CCl_4 + O(^1D)$** | **$\rightarrow 3 \cdot Cl + ClO$** | $2.61 \cdot 10^{-10}$ | JPL2015 |
| **$CF_2ClCFCl_2 + O(^1D)$** | **$\rightarrow 2 \cdot Cl + ClO$** | $2.09 \cdot 10^{-10}$ | JPL2015 |
| **$CF_2ClCFCl_2 + OH$** | **$\rightarrow 2 \cdot Cl + ClO$** | $2.32 \cdot 10^{-10}$ | JPL2015 |
| **$CHF_2Cl + OH$** | **$\rightarrow Cl + H_2O$** | $9.20 \cdot 10^{-13} \cdot \exp(-1560/T)$ | JPL2015 |
| **$CHF_2Cl + O(^1D)$** | **$\rightarrow ClO$** | $7.65 \cdot 10^{-11}$ | JPL2015 |





| $CH_3CCl_3 + OH$ | $\rightarrow COCl_2 + Cl + H_2O$ | $1.64 \cdot 10^{-12} \cdot \exp(-1520/T)$ | JPL2015 |
|---|---|---|---|
| $CH_3CCl_3 + O(^1D)$ | $\rightarrow 2 \cdot Cl + ClO$ | $2.93 \cdot 10^{-10}$ | JPL2015 |
| $CF_3Br + OH$ | $\rightarrow Br$ | $1.00 \cdot 10^{-12} \cdot \exp(-3600/T)$ | JPL2015 |
| $CF_3Br + O(^1D)$ | $\rightarrow BrO$ | $4.50 \cdot 10^{-11}$ | JPL2015 |
| $CH_2Br_2 + OH$ | $\rightarrow 2 \cdot Br + H_2O$ | $2.00 \cdot 10^{-12} \cdot \exp(-840/T)$ | JPL2015 |
| $CH_2Br_2 + O(^1D)$ | $\rightarrow 2 \cdot Br + OH$ | $2.57 \cdot 10^{-10}$ | JPL2015 |
| $CH_2Br_2 + Cl$ | $\rightarrow 2 \cdot Br + HCl$ | $6.30 \cdot 10^{-12} \cdot \exp(-800/T)$ | JPL2015 |
| $CHBr_3 + OH$ | $\rightarrow 3 \cdot Br + H_2O$ | $9.00 \cdot 10^{-13} \cdot \exp(-360/T)$ | JPL2015 |
| $CHBr_3 + O(^1D)$ | $\rightarrow 3 \cdot Br + OH$ | $4.62 \cdot 10^{-10}$ | JPL2015 |
| $CHBr_3 + Cl$ | $\rightarrow 3 \cdot Br + HCl$ | $4.85 \cdot 10^{-12} \cdot \exp(-850/T)$ | JPL2015 |
| $CH_2BrCl + OH$ | $\rightarrow Br + Cl + H_2O$ | $2.10 \cdot 10^{-12} \cdot \exp(-880/T)$ | JPL2015 |
| $CH_2BrCl + Cl$ | $\rightarrow Br + Cl + HCl$ | $6.80 \cdot 10^{-12} \cdot \exp(-870/T)$ | JPL2015 |
| $CHBr_2Cl + OH$ | $\rightarrow 2 \cdot Br + Cl + H_2O$ | $9.00 \cdot 10^{-13} \cdot \exp(-420/T)$ | JPL2015 |
| $CHBrCl_2 + OH$ | $\rightarrow Br + 2 \cdot Cl + H_2O$ | $9.40 \cdot 10^{-13} \cdot \exp(-510/T)$ | JPL2015 |
| $CF_2ClCF_2Cl + O(^1D)$ | $\rightarrow ClO + Cl$ | $1.17 \cdot 10^{-10} \cdot \exp(25/T)$ | JPL2015 |
| $CF_2ClCF_3 + O(^1D)$ | $\rightarrow ClO$ | $4.64 \cdot 10^{-11} \cdot \exp(30/T)$ | JPL2015 |
| $CH_3CFCl_2 + O(^1D)$ | $\rightarrow ClO + Cl$ | $1.79 \cdot 10^{-10}$ | JPL2015 |
| $CH_3CF_2Cl + O(^1D)$ | $\rightarrow ClO$ | $1.30 \cdot 10^{-10}$ | JPL2015 |
| $CF_2ClCF_2Cl + OH$ | $\rightarrow ClO + Cl$ | $1.00 \cdot 10^{-11} \cdot \exp(-6200/T)$ | JPL2015 |
| $CF_2ClCF_3 + OH$ | $\rightarrow ClO$ | $1.00 \cdot 10^{-11} \cdot \exp(-6200/T)$ | JPL2015 |
| $CH_3CFCl_2 + OH$ | $\rightarrow 2 \cdot Cl + H_2O$ | $1.25 \cdot 10^{-12} \cdot \exp(-1600/T)$ | JPL2015 |
| $CH_3CF_2Cl + OH$ | $\rightarrow Cl + H_2O$ | $1.30 \cdot 10^{-12} \cdot \exp(-1170/T)$ | JPL2015 |



| | | | |
|---|---|---|---|
| **$CH_2Cl_2$ + OH** | **→ $CHCl_2O_2$ + $H_2O$** | $1.92 \cdot 10^{-12} \cdot \exp(-880/T)$ | JPL2015 |
| **$CHCl_3$ + OH** | **→ $COCl_2$ + Cl + $H_2O$** | $2.20 \cdot 10^{-12} \cdot \exp(-920/T)$ | JPL2015 |
| **$C_2H_4Cl_2$ + OH** | **→ 2·Cl + $H_2O$** | $1.14 \cdot 10^{-11} \cdot \exp(-1150/T)$ | JPL2015 |
| **$CH_2Cl_2$ + Cl** | **→ $CHCl_2O_2$ + HCl** | $7.40 \cdot 10^{-12} \cdot \exp(-910/T)$ | JPL2015 |
| **$CHCl_3$ + Cl** | **→ $COCl_2$ + Cl + HCl** | $3.30 \cdot 10^{-12} \cdot \exp(-990/T)$ | JPL2015 |
| **$C_2H_4Cl_2$ + Cl** | **→ 2·Cl + HCl** | $1.30 \cdot 10^{-12}$ | Wallington et al. (1996) |
| **$C_2Cl_4$ + OH** | **→ 0.47·$COCl_2$ + 3.06·Cl** | $4.70 \cdot 10^{-12} \cdot \exp(-990/T)$ | JPL2015 |
| **$CHCl_2O_2$ + NO** | **→ 2·Cl + $NO_2$ + CO + $HO_2$** | $4.05 \cdot 10^{-12} \cdot \exp(360/T)$ | MCM3.1 |
| **$CHCl_2O_2$ + $NO_3$** | **→ 2·Cl + $NO_2$ + CO + $HO_2$** | $2.30 \cdot 10^{-12}$ | MCM3.1 |
| **$CHCl_2O_2$ + $HO_2$** | **→ $COCl_2$ + $H_2O$** | $3.92 \cdot 10^{-13} \cdot \exp(700/T)$ | MCM3.1 |
| **$CHCl_2O_2$ + $HO_2$** | **→ $HO_2$ + CO + Cl + HOCl** | $1.68 \cdot 10^{-13} \cdot \exp(700/T)$ | MCM3.1 |
| **$CHCl_2O_2$ + $CH_3O_2$** | **→ 2·Cl + 2·$HO_2$ + CO + HCHO** | $1.20 \cdot 10^{-12}$ | MCM3.1 |
| **$CHCl_2O_2$ + $CH_3O_2$** | **→ $COCl_2$ + HCHO + $HO_2$** | $0.80 \cdot 10^{-12}$ | MCM3.1 |
| **$Br_2$ + OH** | **→ Br + HOBr** | $2.10 \cdot 10^{-11} \cdot \exp(240/T)$ | JPL2015 |
| **$Cl_2$ + OH** | **→ Cl + HOCl** | $2.60 \cdot 10^{-12} \cdot \exp(-1100/T)$ | JPL2015 |
| **HBr + OH** | **→ Br + $H_2O$** | $5.50 \cdot 10^{-12} \cdot \exp(200/T)$ | JPL2015 |
| **$ClNO_2$ + OH** | **→ HOCl + $NO_2$** | $2.40 \cdot 10^{-12} \cdot \exp(-1250/T)$ | JPL2015 |
| **$COCl_2$ + O($^1$D)** | **→ CO + ClO + Cl** | $1.76 \cdot 10^{-10} \cdot \exp(30/T)$ | JPL2015 |
| **$Cl_2$ + O($^1$D)** | **→ Cl + ClO** | $2.03 \cdot 10^{-10}$ | JPL2015 |

**Table 3. Summary of bimolecular reactions of atmospheric halogens included in DEST. Reactions and/or channels absent in the standard StratTrop scheme are highlighted in a bold font and underlined. Unless stated otherwise, all rate constants have been updated to JPL (2015).**






| Reactant | Products |
|---|---|
| BrCl | → Br + Cl |
| BrO | → Br + ($O^3P$) |
| BrONO$_2$ | → Br + NO$_3$<br>→ Br O + NO$_2$ |
| OClO | → O($^3$P) + ClO |
| HOBr | → OH + Br |
| ClONO$_2$ | → Cl + NO$_3$<br>→ ClO + NO$_2$ |
| HCl | → H + Cl |
| HOCl | → OH + Cl |
| Cl$_2$O$_2$ | → 2·Cl + O$_2$ |
| CFCl$_3$ | → 3·Cl |
| CF$_2$Cl$_2$ | → 2·Cl |
| CH$_3$Br | → Br + H |
| **CH$_3$Cl** | **→ Cl + H** |
| **CF$_2$ClBr** | **→ Cl + Br** |
| **CCl$_4$** | **→ COCl$_2$ + 2·Cl** |
| **CF$_2$ClCFCl$_2$** | **→ 3·Cl** |
| **CHF$_2$Cl** | **→ Cl** |
| **CH$_3$CCl$_3$** | **→ 3·Cl** |
| **CF$_3$Br** | **→ Br** |
| **CH$_2$Br$_2$** | **→ 2·Br** |





| | |
|---|---|
| **$\underline{CHBr_3}$** | **$\rightarrow 3 \cdot Br$** |
| **$\underline{CF_2ClCF_2Cl}$** | **$\rightarrow 2 \cdot Cl$** |
| **$\underline{CF_2ClCF_3}$** | **$\rightarrow Cl$** |
| **$\underline{CH_3CFCl_2}$** | **$\rightarrow 2 \cdot Cl$** |
| **$\underline{CH_3CF_2Cl}$** | **$\rightarrow Cl$** |
| **$\underline{CH_2Cl_2}$** | **$\rightarrow 2 \cdot Cl$** |
| **$\underline{CHCl_3}$** | **$\rightarrow 3 \cdot Cl$** |
| **$\underline{C_2H_4Cl_2}$** | **$\rightarrow 2 \cdot Cl$** |
| **$\underline{C_2Cl_4}$** | **$\rightarrow 4 \cdot Cl$** |
| **$\underline{COCl_2}$** | **$\rightarrow 2 \cdot Cl + CO$** |
| **$\underline{ClNO_2}$** | **$\rightarrow Cl + NO_2$** |
| **$\underline{CHCl_2O_2}$** | **$\rightarrow Cl + ClO + OH$** |
| **$\underline{CH_2BrCl}$** | **$\rightarrow Br + Cl$** |
| **$\underline{CHBr_2Cl}$** | **$\rightarrow 2 \cdot Br + Cl$** |
| **$\underline{CHBrCl_2}$** | **$\rightarrow Br + 2 \cdot Cl$** |
| **$\underline{Cl_2}$** | **$\rightarrow 2 \cdot Cl$** |
| **$\underline{Br_2}$** | **$\rightarrow 2 \cdot Br$** |

**Table 4. Summary of photolysis reactions of atmospheric halogens included in DEST. Reactions absent in the standard StratTrop scheme are highlighted in a bold font and underlined.**






| Reactants | Products | Source |
|---|---|---|
| $BrO + NO_2 + M$ | $\rightarrow BrONO_2 + M$ | JPL2011* |
| $ClO + ClO + M$ | $\rightarrow Cl_2O_2 + M$ | JPL2011 |
| $Cl_2O_2 + M$ | $\rightarrow ClO + ClO + M$ | IUPAC2007** |
| $ClO + NO_2 + M$ | $\rightarrow ClONO_2 + M$ | JPL2011*** |
| **$\underline{Cl + C_2Cl_4 + M}$** | **$\underline{\rightarrow 5 \cdot Cl + M}$** | JPL2015 |
| **$\underline{Cl + NO_2 + M}$** | **$\underline{\rightarrow ClNO_2 + M}$** | JPL2015 |

**Table 5. Summary of termolecular reactions of atmospheric halogens included in DEST. Reactions absent in the standard StratTrop scheme are highlighted in a bold font and underlined. *JPL2011 data for the low pressure limit, but with $6.9 \times 10^{-12}$ used for the high pressure limit. **IUPAC2007 data for the low pressure limit, with $1.8 \cdot 10^{14} \exp(-7690/T)$ used for the high pressure limit. *** JPL2011 data for the low pressure limit, but with $1.5 \times 10^{-11}$ used for the high pressure limit.**






| Reactants | Products | NAT | Ice | Aerosols |
|---|---|---|---|---|
| $ClONO_2 + H_2O$ | $\rightarrow HOCl + HONO_2$ | X | X | X |
| $ClONO_2 + HCl$ | $\rightarrow \mathbf{\underline{Cl_2}} + HONO_2$ | X | X | **X** |
| $HOCl + HCl$ | $\rightarrow \mathbf{\underline{Cl_2}} + H_2O$ | X | X | X |
| $N_2O_5 + H_2O$ | $\rightarrow 2 \cdot HONO_2$ | X | X | X |
| $N_2O_5 + HCl$ | $\rightarrow \mathbf{\underline{ClNO_2}} + HONO_2$ | X | X | |
| $ClONO_2 + HBr$ | $\rightarrow BrCl + HONO_2$ | **X** | **X** | |
| $HOCl + HBr$ | $\rightarrow BrCl + H_2O$ | | **X** | |
| $HOBr + HCl$ | $\rightarrow BrCl + H_2O$ | | **X** | |
| $BrONO_2 + HCl$ | $\rightarrow BrCl + HONO_2$ | | **X** | **X** |
| $BrONO_2 + H_2O$ | $\rightarrow HOBr + HONO_2$ | | **X** | **X** |
| $HOBr + HBr$ | $\rightarrow \mathbf{\underline{Br_2}} + H_2O$ | | **X** | |
| $BrONO_2 + HBr$ | $\rightarrow \mathbf{\underline{Br_2}} + HONO_2$ | | **X** | |
| $N_2O_5 + HBr$ | $\rightarrow Br + NO_2 + HONO_2$ | **X** | | |

**Table 6. Summary of the heterogeneous reactions of halogen species in DEST. Columns 3-5 indicate different reaction surfaces (NAT PSCs, ice PSCs, and sulfate aerosols, respectively), and 'X' denotes that a reaction occurs on a given surface in the model. Different product species to those in the standard StratTrop scheme and/or new reaction surfaces are highlighted in a bold font and**
**underlined. See Dennison et al. (2019) for the corresponding uptake coefficients.**