# Peer review of "Description and evaluation of the new UM-UKCA (vn11.0) Double Extended Stratospheric-Tropospheric (DEST vn1.0) scheme for comprehensive modelling of halogen chemistry in the stratosphere."

_Geoscientific Model Development, 2022_

## Author Response (AR1)

**RESPONSE TO REVIEWER #1**

The paper provides a description and an evaluation of the new version of the extended stratospheric-tropospheric scheme in the UM-UKCA model (DEST). The main updates are an explicit treatment of 14 of the long-lived ODS, the inclusion of bromine VSLS emissions and chemistry and the inclusion of chlorine VSLS emissions/lower boundary conditions and chemistry.

The paper fits well within the scope of the journal. It is well written and well organized. The strategy for the evaluation of the changes linked to the new developments is clear. However, the paper could be improved by providing more justification and a more in-depth analysis on a few points as detailed below.

We thank the reviewer for the positive review and helpful comments that have improved the manuscript. We address the individual comments below in blue.

**Major comments**

L210. The 3-member ensemble uses time-varying Cl-VSLS LBCs. Why Cl-VSLS and/or Br-VSLS ? Is the variability more important for Cl-VSLS ? Are there larger uncertainties on Cl-VSLS LBC/emissions than on Br-VSLS LBC/emissions ?

The reviewer is correct: while a fair number of continuous station-based measurements exists for Cl-VSLS (i.e. from the NOAA and AGAGE networks), allowing a reasonable estimate of their long-term trends, the corresponding Br-VSLS measurements are sparse. To our best knowledge, NOAA and AGAGE do not routinely report measurements of Br-VSLS. And whilst a number of aircraft measurement campaigns have reported measurements of Br-VSLS, these are sparse in space/time and, thus, not well suited for diagnosing long-term trends (should such a trend exist). In addition, the much lower concentrations of certain Br-VSLS, thus detectability issues, constitute an additional challenge in confident estimation of their long-term trends.

L214-L222. Two problems were identified in the code used for the DEST 20-year long simulations run in the time slice year 2000. The authors acknowledge that these issues are important. They also say that they should not have a dominant impact on the evaluation. This statement needs to be supported by a more detailed argumentation. Regarding the photolysis cross-section of CFC-113, how is the update expected to modify the stratospheric chlorine budget (Cly) ? For COCl2 + O(1D), it would be interesting to quantify the loss of chlorine associated to this problem, possibly based on a test experiment over one year.

We agree with the reviewer that this issue requires a more detailed argumentation. We have now added Figure S1 to the supplement that diagnoses the cumulative impact of these two issues for the 2010-2019 mean conditions, in a set of transient free-running simulations both with and without Cl-VSLS included (as used in Bednarz et al. 2022). We have also added the following text to Section 4.1.: "As shown in Fig. S1a in the Supplement, the cumulative effect of the two issues under the mean 2010-2019 conditions is to underestimate the total chlorine levels by ~1-2 ppt in the stratosphere, which constitutes a very small fraction of the total stratospheric chlorine content (~3.3-3.4 ppb). In the absence of Cl-VSLS, on the other hand,

the cumulative effect changes sign, leading in turn to a similarly small overestimation of the stratospheric chlorine levels by ~1-2 ppt (Fig. S1b)."

L238-L240. In Fig. 1b, there are lower bromine levels in DEST compared to StratTrop in the southern hemisphere (Fig 1b). Since there are Br-VSLS included in DEST and not in StratTrop, one would think that the bromine levels in the troposphere would be greater in DEST everywhere. What could be the cause of this negative anomaly in DEST with respect to StratTrop ?

As discussed in Section 3.2, while the standard StratTrop UM-UKCA scheme does not include explicit representation of Br-VSLS, their contribution to the total bromine budget is approximated by adding an extra 5 ppt bromine to the LBCs of CH3Br. Thus, the simulated differences in atmospheric bromine levels are the result of not only differences in magnitudes and horizontal distribution of its lower boundary conditions (the surface bromine concentrations simulated in the SH high latitudes are actually lower in DEST than in StratTrop, see Fig. 1b) but also the lifetime of species bromine is present in (in particular CH3Br vs Br-VSLS). We have added this discussion to the manuscript.

Section 4.2.1. To support the analysis of the results discussed in this section, would it be possible to have an estimation of the differences in the LBC concentrations for the ODS between DEST and StratTrop ? This will help to know if the ODS levels are expected to be lower or higher in DEST compared to StratTrop.

We have added this information to the text.

Sections 4.2.2 and 4.2.3 shows that the changes of stratospheric ozone affect stratospheric climate (temperature, H2O, age of air). Apart from the effect of chlorine and bromine emissions, is there any significant impact on PSCs between the DEST and the StratTrop simulations?

Yes, and this is discussed at the bottom of Section 4.2.1: "We find markedly higher levels of reactive chlorine in DEST in the high latitude lower stratosphere in both hemispheres […]. These yearly mean values correspond to accelerated heterogenous reactions on PSCs and aerosols inside the polar vortices in winter and spring. The response reflects the combined impact of the updates to the heterogeneous halogen reactions (Section 3.4), as already discussed in Dennison et al. (2019), and the increase in total stratospheric chlorine in DEST compared to StratTrop (Fig. 1a) as the result of including Cl-VSLS."

Section 4.3. Comparison with the ACE-FTS products. The uncertainty on the ACE-FTS stratospheric ozone concentrations is mentioned L315-317. More generally, it would be useful to know what is the uncertainty on each of the ACE-FTS products used to assess the DEST simulations. This would give an indication of the ability of the DEST simulation to be within these uncertainty ranges.

We agree with the review and have added the associated ACE-FTS uncertainties as Fig. S2 in the Supplement, and we refer to this in the analysis of DEST performance in Section 4.3 as well as in the caption to Figure 9.

Bromine monoxide is also a compound of interest that can be estimated from satellites observations and that shows differences between DEST and StratTrop. I would recommend looking for such data to evaluate the DEST simulation.

We agree it would be useful to compare DEST with satellite observations of BrO. However, BrO measurements are not available from the ACE-FTS satellite dataset used here and a thorough comparison would need careful curation and analysis of multiple satellite datasets, along with careful model sampling, for a robust analysis. As such, we believe this beyond the scope of the present manuscript, but will be the focus of future work in an upcoming study.

**Minor comments**

L248. "differences in in" à "differences in"

Corrected.

L254-L256. Could you add a short explanation on how the heterogeneous bromine reactions are linked to BrO?

We have now modified that part to read:

"For bromine, DEST shows an increase in BrO in the tropical tropopause layer (TTL) and upper troposphere and lower stratosphere (UTLS), and a decrease in BrO above (Fig. 4f); this is thus qualitatively similar to what was found for HCl. The BrO increase is related to the higher total bromine levels simulated in the lowermost stratosphere as the result of including Br-VSLS (as discussed above) as well as to the inclusion of heterogenous bromine reactions (Section 3.4) that increase the BrO to total inorganic bromine ($Br_y$) ratio through the conversion of bromine reservoirs (e.g. HBr). Previous box model analysis highlighted a strong sensitivity of the $BrO/Br_y$ ratio in the TTL to the aerosol and ice surface area density (e.g. Koening et al., 2017)"

L270. I think the reference to Fig. 4 should be Fig. 3c and the reference to Fig3f and Fig. 3c should be Fig 4f and Fig. 4c

That's right, apologies, now corrected.

**RESPONSE TO REVIEWER #2**

This paper describes an extension of the standard chemistry scheme of the UM-UKCA chemistry-climate model and a detailed evaluation of the performances of the new chemistry scheme (called DEST) using comparisons of present-day model simulations against multiple observational datasets, notably satellite data. The model updates are an explicit treatment of key long-lived ozone-depleting substances, of bromine-containing very short-lived species emissions and chemistry, of chlorine-containing very short-lived species emissions. The scheme also includes additional inorganic halogen tracers, and changes to the photolysis, gas-phase and heterogeneous reaction rates. The paper is well written, clear, and a valuable contribution to the stratospheric ozone numerical modelling community. The new model is better suited for studies of stratospheric ozone depletion/recovery associated with both controlled long-lived ozone-depleting substances and, more importantly, uncontrolled very short-lived halogen substances. Its scope fits perfectly with those of GMD model description. Therefore, I recommend publication with very minor comments that the authors may consider.

We thank the reviewer for the positive review and helpful comments that have improved the manuscript. We address the individual comments below in blue.

-p3, l76: …chemistry-climate model…

Added.

-p3, l84: why 'double'?

As discussed in Section 3.1., an earlier UKCA chemistry scheme version exists that included explicit representation of the long-lived ODSs; that scheme is called 'Extended Chemistry of the Stratosphere (CheS+)'. As such our scheme builds on and expands the old scheme by not only including explicit representation of the long-lived ODSs (with even more species) but also the representation of Br-VSLS and Cl-VSLS chemistry. Hence the 'Double Extended'.

We have now spelled out the name of the old CheS+ scheme in the text.

-p4, l110: add '(Lower Boundary Conditions)'

Added.

-p7, l194: It could be interesting for a follow-on study to rerun the time-varying simulations but forced with meteorological analyses. This alternative set-up could make the testing of the chemistry scheme simpler, less dependent on the model dynamical response to chemical composition changes.

We agree with the reviewer. In fact, a nudged DEST simulation was used in our earlier study (Bednarz et al., 2022). We have now added the corresponding analysis of that simulation as a new Figure 10 and extra text in Section 4.3 into the manuscript.

-p7, l216: replace , by .

Corrected.

-p8, l248: remove one 'in'.

Corrected.

-p9, l257: ?? rephrase

Corrected.

-p9, l262: add 'zonal mean' and define reactive chlorine.

Done.

-p9, l283: 'consequence...stability'. not sure to have understood this comment.

We have now clarified this in text.

-p10, l293: Perhaps, the authors should highlight the differences between the analysis/objective by Bernarz et al, 2022 and the present one.

This is a good idea – we have added this to the manuscript.

- p10, l307: As shown by Bernarz et al, 2022, the inclusion...

We have now rephrased this.

-p11, l330: the too cold/persistent Antarctic vortex bias is common to several chemistry-climate models.

The reviewer raises an important point – we have added this to the text.

-p11, l336: It is not totally surprising. Dynamical parametrisations (e.g. orographic and non-orographic waves) of chemistry-climate models are tuned in order to reproduce climatologies. When a scheme, here the chemistry scheme, is changed/improved, this tends to degrade initially the model performances. Dynamical model parameters might need to be tuned again with the new chemistry scheme.

The reviewer raises an important point – we have added this to the text.

---

## Author Response (AR2)

**Authors response**

We apologize for not addressing this comment sufficiently before. This was caused by the confusion on our part as to whether the Reviewer asks about the impacts from or on the PSC; we have previously mistakenly answered the former question. Below we now address the question whether there are any differences in the simulated PSC fields between the DEST and StratTrop simulations:

We have now considered the differences in the simulated PSC fields between the timeslice year 2000 DEST and StratTrop simulations. Unfortunately, we have not outputted the diagnostics required to look at type I NAT PSCs in one of the simulations, and so we could only examine changes in the type II ice PSCs. As shown in Figure R1 and the right column of Figure R2, we do not find substantial and/or consistent changes in the simulated concentrations of ice PSCs in our runs. This arises partially because of the seasonality in the associated differences in the polar lower stratospheric temperatures and water vapour (Left and middle columns in Figure R2).

Given the above, we have not modified the manuscript compared to the previous round of revisions.

[Figure]

**Figure R1.** Shading: Annual mean difference [%] in simulated mass fraction of ice cloud in air between DEST and StratTrop for the year 2000 conditions. Hatching denotes regions where the difference is not statistically significant, here taken as being lower than ±2 standard errors. Contours show the corresponding values in DEST for reference [kg-ice/kg-air; only the 0.01, 0.1, 1, and 10 kg-ice/kg-air contours shown].

[Figure]

**Figure R2.** Shading: Seasonal mean differences in (a) temperature [K], (b) specific humidity [%], and (c) mass fraction of ice cloud in air [%] between DEST and StratTrop for the year 2000 conditions. Hatching denotes regions where the difference is not statistically significant, here taken as being lower than ±2 standard errors. Contours show the corresponding values in DEST for reference.